# Designing Infant Mattresses Tailored to Developmental Sleep Characteristics: A Comprehensive Review

**DOI:** 10.3390/clockssleep7040070

**Published:** 2025-12-08

**Authors:** Yasunori Oka, Akiko Tange, Yuki Maeda

**Affiliations:** 1Center for Sleep Medicine, Ehime University Hospital, Toon 791-0295, Japan; okasleep@gmail.com; 2Unicharm Corporation, Kanonji 769-1602, Japan; yuki-maeda@unicharm.com; 3Graduate School of Humanities and Social Sciences, Hiroshima University, Higashi-Hiroshima 739-8524, Japan

**Keywords:** infant, sleep environment, mattress design, SUID

## Abstract

This paper reviews existing research on infant mattress design to promote safe and comfortable sleep and proposes evidence-based design recommendations. Focusing on safety related to Sudden Unexpected Infant Death (SUID) and comfort associated with infant development and thermoregulation, we examine mattress firmness, pressure distribution, breathability, and thermal properties. Since infants have difficulty turning over and possess immature thermoregulatory functions, mattress characteristics directly influence sleep quality and safety. Based on international studies, we clarify the requirements for infant mattresses and provide insights into future product development and evaluation standards.

## 1. Introduction

Infancy represents a critical stage in human development, during which the sleep environment plays a pivotal role in supporting physical growth, neurological maturation, and overall safety [1]. As infants typically spend the majority of their day asleep, the quality and safety of their sleep environments have a profound impact on their health and developmental outcomes.

In recent years, increasing public concern has been directed toward sleep-related risks in infancy, particularly Sudden Unexpected Infant Death (SUID). According to a national epidemiological survey in the United States, approximately 3400 infant deaths occur annually due to SUID, encompassing Sudden Infant Death Syndrome (SIDS) and suffocation, with many cases attributed to unsafe sleep environments [2]. In response, pediatric societies and health organizations worldwide have issued guidelines to promote safer sleep practices. While these efforts have contributed to a decline in SUID incidence, a substantial number of cases remain unprevented, underscoring the need for scientifically grounded approaches to bedding design and safety evaluation.

Concurrently, there is growing recognition of the importance of sleep comfort in infancy. Sleep disturbances—such as frequent awakenings and difficulty initiating sleep—are increasingly understood to be influenced not only by individual variability but also by bedding and environmental factors. A large-scale web-based survey conducted by Mindell et al. [3], involving 29,287 infants and toddlers aged range from birth to 36 months across 17 countries, revealed that bedding conditions and sleep habits significantly affect nighttime awakenings and sleep onset latency. These findings highlight the necessity of considering regional and cultural differences when standardizing infant sleep environments. Moreover, the immature thermoregulatory systems of infants must be taken into account. Due to their limited ability to regulate body temperature and reposition themselves during sleep, mattress breathability and thermal properties can significantly influence sleep quality [4].

Further evidence from a retrospective study by Schnitzer et al. [5], using 2005 to 2008 data from 9 US states to assess 3136 sleep-related SUIDs, identified the use of soft mattresses and adult bedding as major contributing factors. These findings reinforce the critical role of mattress design in ensuring infant sleep safety.

Given this context, infant mattress design must move beyond traditional, subjective notions such as “soft and cozy,” and instead pursue a balance between safety and comfort based on empirical evidence. This necessitates the clear definition of design parameters aligned with infants’ physiological and behavioral characteristics, and their integration into product development and evaluation frameworks.

This paper aims to contribute to the creation of safe and comfortable sleep environments for infants, tailored to developmental sleep characteristics, by reviewing and synthesizing existing research from international sources. Specifically, it examines how physical mattress attributes—such as firmness, pressure distribution, breathability, and thermal properties—interact with infant physiology and sleep behavior and proposes evidence-based design recommendations. SIDS predominantly occurs within the first year of life (infancy). Therefore, this study will focus on infants during this period, characterized by immature physiological and physical functions. However, research specifically targeting infants within the first year of life is limited. Consequently, to supplement the deficient knowledge base and extend the findings to infants, this review will also include literature on adults, children, and preterm infants. Furthermore, specialized knowledge tends not to fully permeate or be reflected in the behavior of caregivers in general households, and especially among secondary caregivers such as grandparents who do not regularly care for infants. Therefore, we will focus on the infant sleep environment in general households and discuss mattresses used in the home setting.

To identify relevant literature for this study, a systematic search was conducted using two databases: Google Scholar and the AI-powered literature search engine, Consensus. First, in Google Scholar, a total of 12 keywords—Infant, mattress, sleep surface, sleep position, body pressure, sleeping environment, SIDS, infant death, breath, blood flow, myoelectric, and skeleton—were used to search for literature, resulting in 31 hits. Next, the search was conducted on Consensus, which focuses on peer-reviewed articles. The search employed four question-based queries deemed highly relevant from a psychophysiological perspective: “Relationship between psycho-physiologically appropriate infant sleeping posture and mattress firmness,” “Relationship between infant psychomotor development and mattress firmness,” “Car seats, posture, and health risks,” and “Mattress, respiration, and posture.” This search yielded 43 hits. The combined total from both search databases was 74 articles (31 from Google Scholar + 43 from Consensus). After conducting a screening process based on the relevance of these 74 articles to the objectives of the current study, a final selection of 30 articles was chosen for use in the discussion of this research.

## 2. Basic Knowledge of Infant Physical Characteristics and Sleep Posture

A thorough understanding of infants’ physical characteristics is essential for designing mattresses that support safe and healthy sleep. Newborns and young infants possess flexible skeletal structures, disproportionately large heads, and underdeveloped muscle strength, which limit their ability to reposition themselves or maintain stable postures during sleep [6]. Consequently, pressure tends to accumulate on specific body regions, necessitating bedding materials with superior pressure distribution capabilities.

The infant skull is highly plastic, and prolonged pressure in a single direction can result in positional cranial deformation. A prospective study conducted in New Zealand examined the relationship between sleep posture and head shape in 100 infants those received a diagnosis of having non-synostotic plagiocephaly, revealing that extended supine sleep significantly increases the risk of cranial deformation [7]. Similarly, a Canadian cohort study involving 440 infants aged 2–3 months reported a 46.6% prevalence of positional cranial deformation, with 63.2% occurring on the right side and 36.0% on the left. The authors attributed this trend to the widespread recommendation of supine sleep for SUID prevention, which inadvertently increases time spent in a fixed position [8]. These findings underscore the importance of mattress surface structures that mitigate pressure on the head.

Priyadarshi et al. [9] conducted a systematic review evaluating the impact of sleep posture (supine vs. non-supine) on health outcomes in healthy infants under one year of age. Although the certainty of evidence was limited, the supine position was associated with a reduced risk of SUID. However, short-term delays in gross motor development and an increased incidence of positional plagiocephaly at six months were observed, with no significant long-term neurodevelopmental effects noted at 18 months.

Conversely, the supine position may offer benefits for sleep quality. An integrative review by Modesto et al. [10], indicated that in the prone position, the infant experienced fewer arousal events, enabling deeper active sleep and a greater amount of quiet sleep. The supine position is associated with a greater amount of active sleep and more arousals, but it does not expose the infant to the risk of SIDS. Additionally, Siddicky et al. [11] in the United States, assessed spinal muscle activity in 22 infants aged 2–6 months across five positions, lying prone, lying supine, held in-arms, held in a baby carrier, and buckled into a car seat, demonstrating that infants exhibited significantly higher erector spinae activity during prone positioning compared to all other tasks.

Based on these insights, the following considerations are essential for infant mattress design.

(1)Providing appropriate elasticity and pressure distribution to prevent localized pressure accumulation.(2)Incorporating surface structures that reduce the risk of cranial deformation.

These elements are not solely for enhancing comfort; they represent critical design requirements directly linked to long-term physical development and sleep safety.

## 3. Research Findings on Mattress Firmness and Pressure Distribution

Infants possess underdeveloped skeletal and muscular systems, which limit their ability to independently adjust sleeping postures. This makes it particularly important to minimize localized pressure on prominent body regions such as the occiput and buttocks, in order to prevent circulatory disturbances and deformation resulting from prolonged compression. Accordingly, effective pressure distribution is a critical design requirement for infant mattresses. Additionally, excessively soft mattresses may allow the infant’s face to sink into the surface, increasing the risk of SUID. Therefore, achieving an optimal balance between firmness and pressure distribution is essential.

While extensive research has been conducted on mattress performance in adults, including its effects on pressure distribution, support, and comfort, such studies offer valuable insights for infant mattress design. For instance, Ren et al. [12] evaluated ten mattress types with varying bedding layer structures in ten adult participants (five males and five females, mean aged of approximately 23 years). Their findings indicated that mattresses with a soft surface layer and progressively firmer lower layers increased the low-pressure contact area by 15.86% in the supine position and 27.15% in the lateral position, while reducing the high-pressure contact area by 64.83% and 47.42%, respectively. Maximum pressure decreased by 17.72% in the supine position and 17.55% in the lateral position, with significant improvements in subjective comfort scores. Since adults naturally shift positions during sleep to restore blood flow in compressed tissues, controlling localized pressure contributes to enhanced sleep comfort.

Rayward et al. [13] employed finite element analysis to investigate the impact of mattress firmness on tissue compression stress in the supine position. Using a model of the pelvic region of a healthy adult male, they found that maximum compressive stress in the sacral area was 18.5 kPa on soft foam and 30.9 kPa on firm foam—a 67% increase. Stress levels also rose by 20% at the ischium, 42% at the lesser trochanter, and 50% at the skin. These results suggest that soft substrates may exhibit nonlinear stress dispersion effects, thereby reducing peak stress in critical regions.

Although research specifically targeting infants remains limited, some notable findings have emerged. Yu et al. [14] assessed the influence of mattress firmness on body pressure distribution in 36 infants aged 0–3 years in the supine position. Their study revealed that mattresses with medium firmness yielded an average maximum occipital pressure of 14.2 kPa, significantly lower than soft (22.8 kPa) and firm (18.5 kPa) mattresses, indicating superior pressure distribution performance.

These findings emphasize the need for standardized, objective methods to define and assess “appropriate firmness” in infant mattresses. Furthermore, material design and evaluation protocols must be developed to ensure a balance between firmness and pressure distribution. Selecting a well-designed mattress not only supports safe sleep posture but also lays the foundation for a comfortable and developmentally supportive sleep environment.

## 4. Perspective on Comfort: Infant Characteristics and Thermal Environment in the Sleep Space

Creating a comfortable sleep environment for infants requires consideration not only of safety but also of physical and physiological comfort. Infants—particularly preterm and newborns—possess immature thermoregulatory systems, making their core body temperature highly sensitive to ambient conditions. Knobel-Dail [15] noted that newborns exhibit reduced heat dissipation capacity compared to adults due to underdeveloped mechanisms of heat loss via skin blood flow. Additionally, they rely heavily on non-shivering thermogenesis through brown adipose tissue for metabolic heat production, which increases both oxygen consumption and metabolic load. As a result, infants tend to maintain a core body temperature approximately 0.3–0.5 °C higher than adults, and this is strongly influenced by the thermal properties of their surrounding environment.

During the early months of life, infants often remain in the same position for extended periods due to limited mobility, leading to localized accumulation of heat and moisture. This condition can negatively impact sleep quality and physiological stability, making the thermal characteristics of the mattress a critical factor in ensuring comfort.

Because infants sleep with a large portion of their body surface in contact with the mattress, they are particularly prone to skin dampness and heat retention. In a study involving adult participants, Li et al. [16] evaluated four types of mattresses and identified optimal thermal comfort zones within the bed microclimate that contributed to improved sleep quality. Mattresses equipped with water circulation systems and 3D airy mesh fabrics outperformed standard latex foam mattresses. The reported comfortable temperature ranges for the contact surface and bed microclimate were 32.3–33.8 °C and 32.8–33.6 °C, respectively. These findings suggest that materials with temperature regulation and breathability can enhance sleep quality and help establish thermal comfort zones.

Although most existing studies focus on adults, emerging research indicates that heat emitted from the infant body can accumulate in bedding, raising both skin and core temperatures and potentially disrupting sleep–wake rhythms. Bedding materials with low breathability or excessive insulation may induce discomfort and arousal responses due to elevated body temperature. Baddock et al. [17] reported that infants who shared a bed exhibited higher skin temperatures—particularly in the lower limbs—and a tendency toward elevated core temperatures compared to those sleeping in cots. Franco et al. [18] found that increased ambient temperature raised the auditory arousal threshold in infants, suggesting that warmer environments may reduce the likelihood of waking, supporting the hypothesis that heat retention in bedding affects arousal mechanisms.

The underdevelopment of sweat gland function further limits heat dissipation in infants. Foster et al. [19] conducted thermal stimulation and acetylcholine injection tests in neonates and found that sweat gland responses were generally absent until approximately 225 days after birth. Due to this immature sweat response, discomfort caused by skin dampness is a significant concern, and the use of materials with excellent moisture absorption and diffusion properties is highly desirable.

From this perspective, the structure and breathability of mattress materials are directly linked to sleep comfort. Materials with high porosity and breathability—such as three-dimensional fiber mesh structures—can prevent the buildup of heat and moisture, thereby improving the sleep microclimate. These materials also offer superior pressure distribution, enabling a balance between breathability, comfort, and safety [20].

Mattresses have also been evaluated in terms of thermal conductivity, CO_2_ accumulation, and emissions of volatile substances. Zamora et al. [21] compared seven types of infant mattresses in Spain. Based on the validation results, the authors proposed design recommendations with defined safety thresholds, which may help reduce the risks of heat retention and rebreathing. Regarding mattress surface materials, Stoltz et al. [22] compared standard vinyl-covered foam mattresses and viscoelastic polyurethane mattresses (VPM) in neonatal intensive care units, assessing nurse evaluations, parental impressions, and infant vital signs over a 72 h period. Although no statistically significant differences were observed, the VPM group showed a slightly greater tendency for weight loss.

In mattress design, it is important to avoid excessive insulation by considering infant skin temperature (approximately 36–37 °C) and the combined bedding temperature (approximately 32–34 °C). Selecting materials with high breathability and moisture absorption can reduce skin dampness and heat retention, potentially improving sleep quality by facilitating sleep onset and reducing nighttime awakenings.

In summary, achieving a comfortable sleep environment for infants requires thermal design grounded in physiological characteristics. It is therefore essential to maintain optimal temperature and humidity conditions within the sleep space. A comprehensive approach that integrates material properties, structural design, breathability, ambient conditions, and interactions with other bedding components is vital.

## 5. Safety Guidelines and Evaluation Methods

Safety guidelines for infant sleep environments have been established by pediatric societies across various countries. Notably, the American Academy of Pediatrics has issued comprehensive recommendations aimed at preventing sleep-related infant deaths [2]. Key recommendations include:(1)Placing infants to sleep on their backs (avoiding prone positions)(2)Using firm, flat mattresses(3)Eliminating soft bedding (e.g., pillows, blankets, stuffed animals)(4)Room-sharing with parents without bed-sharing.

These guidelines are designed to mitigate risks such as airway obstruction caused by facial sinking into bedding and impaired arousal responses due to excessive thermal insulation.

Experimental research has further validated these concerns. Wang et al. [23] placed infants in a supine position on mattresses with inclines of 0°, 15°, and 30°, and measured muscle activity in infants aged 1–3 months (*n* = 15). The results showed that increased incline led to significantly greater muscle strain in the neck and lower back. At a 30° incline, the neck flexion angle exceeded clinical safety thresholds, indicating an elevated risk of airway obstruction.

Observational studies have also highlighted the dangers of improper sleep posture. Plumptre et al. [24] conducted a systematic review to examine the evidence regarding the potential involvement of car seats in SIDS. It has been shown that oxygen saturation decreases in infants seated in car seats under simulated travel conditions. This finding is consistent with another report indicating that 48% of car seats–related deaths are due to positional asphyxia [25]. In addition, prolonged stays in car seats have also been reported to increase the risk of airway obstruction. These findings reinforce the importance of avoiding inclined or sinking postures that may compromise respiratory function.

Regarding bedding characteristics, several studies have quantitatively assessed facial sinking. Gillani et al. [26] developed a device to measure mattress firmness and evaluated 17 infant and adult sleep products available in the U.S. market. Only 24% met the safety threshold value of sinking depth, with softer products posing a higher risk of facial immersion. Somers [27] proposed a do-it-yourself mattress firmness test for home use. Through experimentation across 34 sleep surfaces of varying firmness, an alternative testing method was identified.

In Japan, Kanetake et al. [28] evaluated five types of infant mattresses using mannequins to measure the half-life of exhaled CO_2_ (t_1_/_2_). Their analysis revealed that new, firm mattresses exhibited shorter t_1_/_2_ values, indicating lower rebreathing risk and supporting the recommendation for firmer sleep surfaces.

A retrospective analysis by Schnitzer et al. [15] of SUID cases further identified bedding softness and improper usage as major contributing factors to sleep-related accidents. Ensuring safety thus requires not only appropriate product design but also education, awareness, and targeted interventions within the home environment.

From a behavioral standpoint, Colson et al. [29] surveyed 3297 mothers across the United States regarding infant sleep posture. While 77.3% reported usually placing their infants in a supine position, only 43.7% consistently did so. The study found that parental attitudes, perceived social norms, and physician guidance were key factors influencing behavioral change, underscoring the importance of educational interventions.

Taken together, the design and dissemination of safe infant mattresses require a comprehensive approach built upon three foundational pillars:(1)Clear safety standards(2)Development of objective evaluation methods(3)Consumer education and awareness

Furthermore, the international standardization of product specifications and safety guidelines remains a critical issue for future policy development and global implementation.

## 6. Proposed Designs of Infants Mattress

Based on the findings presented in this review, the following design proposals are recommended for infant mattresses. These suggestions integrate both comfort and safety, tailored to developmental sleep characteristics, grounded in scientific evidence, and are intended to serve as a foundation for future product development and evaluation standards.

### 6.1. Pressure Distribution Performance

Infants often remain in the same position for extended periods, making it essential to minimize concentrated pressure on specific body regions. As demonstrated by Yu et al. [14], materials that combine moderate elasticity with adequate support can effectively reduce physical stress caused by compression. Mattress designs incorporating such materials are therefore strongly recommended.

### 6.2. Thermal Regulation Function

Due to their relatively higher body temperatures and limited thermoregulatory capacity, infants are prone to heat retention and excessive sweating. Fang et al. [20] reported that highly breathable materials—such as three-dimensional fiber mesh structures—can prevent the accumulation of heat and moisture, thereby creating a more comfortable sleep microclimate. In selecting materials, attention should also be given to moisture absorption and thermal conductivity to enhance thermal comfort.

### 6.3. Ensuring Safety

To mitigate the risk of suffocation caused by facial sinking or limited mobility, infant mattresses must be designed with an appropriate level of firmness. Excessive softness has been identified as a potential risk factor for SUID. Referring to the standards proposed by Gillani et al. [26] and Somers [27], the adoption of objective firmness evaluation metrics is highly desirable to ensure safety.

### 6.4. Standardization of Evaluation

Performance indicators such as mattress firmness, breathability, and moisture absorption are often difficult for consumers to interpret. Therefore, the introduction of user-friendly indicators and labeling systems—such as the simple home evaluation method for mattress firmness proposed by [27]—is necessary to enhance transparency and facilitate informed decision-making.

Figure 1 illustrates the developmental characteristics associated with the infant sleep environment. In addition, Figure 2 presents the proposed design elements for infant mattresses. These proposals should not only serve as benchmarks for product development but also be incorporated into educational and awareness-raising initiatives related to infant sleep environments. A flexible design approach that accommodates diverse user needs and living conditions will significantly contribute to the health and safety of infants.

## 7. Future Challenges and Research Directions

Although research on infant bedding has expanded in recent years, clinical evidence and understanding of long-term effects remain limited. In particular, to elucidate the relationship between thermal environments, sleep quality, and developmental outcomes, further longitudinal and multi-center empirical studies are urgently needed.

To date, most studies have focused on individual mattress attributes such as firmness and breathability [14,20]. However, it is essential to evaluate the effectiveness of integrated designs that consider the total sleep environment. Such assessments should encompass not only the mattress itself but also sleepwear, blankets, room temperature, and humidity, in order to understand the comprehensive impact of the sleep setting on infant health and development.

Specifically, accidental suffocation or strangulation in bed is one of the most common causes of SUID [2]; therefore, educating caregivers is considered crucial regarding the importance of keeping blankets, sleepwear, and toys with low breathability that could cover an infant’s face, or those with cord-like parts that present a strangulation risk, out of the bed. Furthermore, a study by Howard et al. in the United States [30] reported that caregivers from high infant mortality risk areas were more likely to have infants sleep in locations other than their own home and were more likely to report that others were caring for the infant during sleep. This suggests that education on safe and comfortable sleep is also necessary for non-parental caregivers, such as grandparents, relatives, friends, and childcare providers.

Regarding product evaluation standards, current methodologies are largely adapted from adult testing protocols and specifications [12,13]. It is imperative to establish evaluation systems that reflect the unique physical and behavioral characteristics of infants. In particular, the development of practical and quantitative testing methods—as well as labeling systems for key parameters such as firmness, breathability, and moisture absorption—will support future product innovation and provide caregivers with reliable information for informed decision-making.

From an international perspective, the harmonization of safety standards and design practices across countries is a pressing issue. In addition to developing environments tailored to regional contexts, collaborative international research and information-sharing frameworks are needed to enhance the global quality and safety of infant bedding.

In summary, the future of research and practical implementation depends on the accumulation of robust clinical evidence, the advancement of product evaluation standards, and a multidisciplinary approach that fosters collaboration among researchers, healthcare professionals, manufacturers, and policymakers.

## 8. Conclusions

In the context of infant sleep environments, the mattress plays a pivotal role in ensuring both comfort and safety. This review examined key design elements—such as firmness, pressure distribution, breathability, and thermal properties—based on the physiological and developmental characteristics of infants, and proposed evidence-based guidelines for mattress design.

From the perspective of comfort, the findings suggest that optimizing the thermal environment for infants with immature thermoregulatory functions, along with ensuring adequate breathability and moisture absorption, can significantly enhance sleep quality. From the perspective of safety, it was confirmed that appropriate standards for mattress firmness and surface structure are essential to reduce the risks associated with SUID, particularly SIDS and suffocation. The importance of simple home evaluation methods and alignment with international safety guidelines was also emphasized.

This review highlights that the requirements for infant mattresses extend beyond individual physical properties and must incorporate multiple dimensions—including comfort, safety, and ease of evaluation—into a cohesive design framework. Future product development should aim to balance these elements, supported by robust evaluation systems that facilitate both implementation and dissemination.

Furthermore, collaboration among caregivers, healthcare professionals, researchers, and manufacturers is essential to promote scientifically grounded information sharing and continuous product improvement. Such efforts will ultimately contribute to the creation of a safer and more comfortable sleep environment for infants worldwide.

## Figures and Tables

**Figure 1 clockssleep-07-00070-f001:**
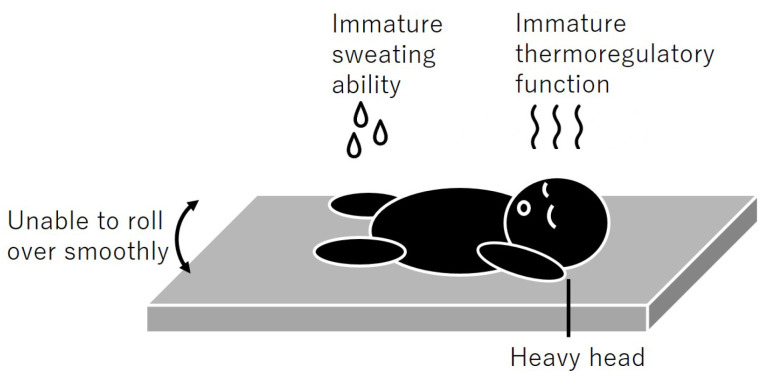
The developmental characteristics associated with the infant sleep environment.

**Figure 2 clockssleep-07-00070-f002:**
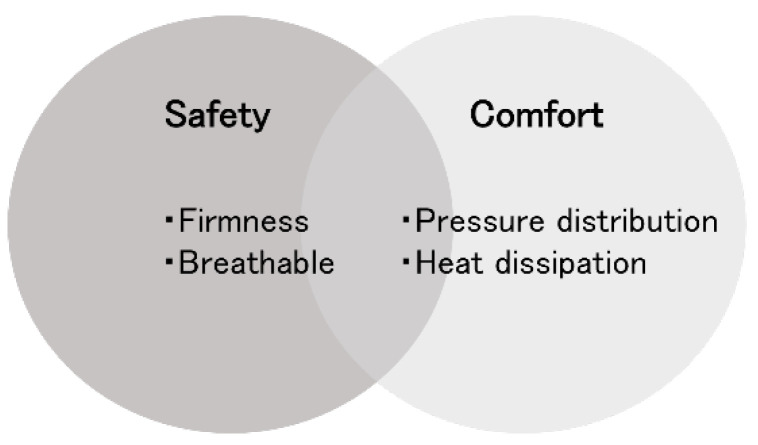
The proposed design elements for infant mattresses.

## Data Availability

No new data were created or analyzed in this study. Data sharing is not applicable to this article.

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
