# Peer review of "Designing Infant Mattresses Tailored to Developmental Sleep Characteristics: A Comprehensive Review"

_2624-5175, 2025, doi:10.3390/clockssleep7040070_

Round 1
Reviewer 1 Report
Comments and Suggestions for Authors
Thank you for the opportunity to review this paper reviewing the parameters and recommendations related to infant mattresses. The comments are intended to improve the clarity of this review.
Were there any methodology used to conduct this search? If not, were there search parameters or constraints placed on this review of literature that should be described? What databases were used? Inclusion and exclusion criteria, keywords etc.
Section 4: The first sentence mentions preterm and newborns. The authors also use the word ‘infant’ throughout. It might be helpful if the author define the terms used to ensure a consistent understanding of how these concepts are applied within the paper. Beyond this point, it may be important to indicate how this work, like evaluating mattress standards, should be considered separately for preterm vs. full term. Full term newborns are different from preterm newborns noting that preterm newborns themselves have several categories of differentiation (late preterm, near term, and all the different low birth weight designations). Also, conceptually should mattresses parameters used in healthcare facilities for infants in the NICU be included or are they different enough from full term newborns that they should be excluded from this review? I would appreciate hearing the justification for including all types of neonates in addition to infants (>30 days post-birth).
This paper talks about mattresses, but it would be helpful to clarify what types of mattresses are the focus of this review? There are such a variety of consumer products for infant sleep, which products do mattress standards apply to? Does it matter where the product is regulated from? For example, there are mattresses for healthcare environments that may be regulated differently from consumer products. Some countries disallow sale of products that aren’t cribs, play-yards or bassinets (i.e., in-bed products). Defining the scope of this review would be helpful to the validity of the work.
p. 5 bottom “This finding is consistent with another report…” Please cite this report.
I see no citations in the section titled “Future challenges and research directions” despite there being evidence that could be cited to back up the statements.
The conclusions talk about the ‘…simple home evaluations methods’ which focuses on the home, yet in other parts of the paper mattresses in NICU incubators were discussed. Again – getting clear on the paper’s focus and application would improve the paper.
Author Response
Dear Reviewer 1
Thank you very much for the constructive and insightful comments and valuable advice on our manuscript. We greatly appreciate the time and effort spent reviewing our work.
We have carefully considered all the points raised by the reviewers and have thoroughly revised the manuscript accordingly.
Please see the attachment." in the box.
Sincerely,
Tange

Reviewer 2 Report
Comments and Suggestions for Authors
This is an important paper highlighting the complexities of understanding the infant sleep environment and interplay with the infant products and ambient environment. With some minor edits, this paper is ready for publications.
Figure 2 is confusing. The text is clear and easily understood. The Figure itself is confusion - is this a Ven Diagram showing where product materials and the environment should overlap? If so, it is unclear what the line means. There may be a clearer way to represent this idea.
Another factor that is getting more attention in the literature, is the idea that knowledge does not predict change in sleep environment. There have been studies showing multiple factors resulting in infants sleeping in unsafe environments despite knowing there is a risk. Comfort and temperature regulation are two of the factors. The introduction and Future Challenges and Research Directions could benefit from brief discussion of this as well. An example of one article is: Howard MB, Jarvis LR, Badolato GM, Parrish BT, Donnelly KA. Variations in Safe Sleep Practices and Beliefs: Knowledge is not Enough. Matern Child Health J. 2022;26(5):1059–1066. There are other articles that could be cited as well.
Author Response
Dear Reviewer 2
Thank you very much for the constructive and insightful comments and valuable advice on our manuscript. We greatly appreciate the time and effort spent reviewing our work.
We have carefully considered all the points raised by the reviewers and have thoroughly revised the manuscript accordingly.
Please see the attachment." in the box.
Sincerely,
Tange
